# Phytoecdysteroids from *Serratula coronata* L. for Psoriatic Skincare

**DOI:** 10.3390/molecules27113471

**Published:** 2022-05-27

**Authors:** Anna Kroma, Mariola Pawlaczyk, Agnieszka Feliczak-Guzik, Maria Urbańska, Dorota Jenerowicz, Agnieszka Seraszek-Jaros, Małgorzata Kikowska, Justyna Gornowicz-Porowska

**Affiliations:** 1Department and Division of Practical Cosmetology and Skin Diseases Prophylaxis, Poznan University of Medicinal Sciences, 3 Rokietnicka St., 60-806 Poznań, Poland; anna.kroma@ump.edu.pl (A.K.); mariolapawlaczyk@ump.edu.pl (M.P.); murbanska@ump.edu.pl (M.U.); justynagornowicz1@poczta.onet.pl (J.G.-P.); 2Department of Chemistry, Adam Mickiewicz University, 8 Uniwersytetu Poznańskiego St., 61-614 Poznań, Poland; agaguzik@amu.edu.pl; 3Department of Dermatology, Poznan University of Medical Sciences, 49 Przybyszewskiego St., 60-356 Poznań, Poland; djenerowicz@ump.edu.pl; 4Department of Bioinformatics and Computational Biology, Poznan University of Medical Sciences, 4 Rokietnicka St., 60-806 Poznań, Poland; agnetpa@gmail.com; 5Laboratory of Pharmaceutical Biology and Plant Biotechnology, Department and Division of Practical Cosmetology and Skin Diseases Prophylaxis, Poznan University of Medicinal Sciences, 3 Rokietnicka St., 60-806 Poznań, Poland

**Keywords:** psoriasis, skin care, *Serratula coronata*

## Abstract

Phytoecdysones from *Serratula coronata* seem to be promising agents for skincare in patients with psoriasis. The aim of the study was to determine the effects of creams containing the extract of *S. coronata* on psoriatic lesions. Creams with different formulas were prepared: 0-Lekobaza^®^, 1-Lekobaza^®^, *S. coronata*, 2-Lekobaza^®^, Salicylic acid, 3-Lekobaza^®^, *S. coronata*, Salicylic acid. After examination of skin penetration and biosafety, the designated cream was applied twice daily for 6 weeks on 72 psoriatic plaques located on elbows or knees. The lesions were assessed at baseline and follow-up of 6 weeks. The lesions area was measured, and severity of scaling, erythema, and infiltration was assessed using a 5-point scale (from 0—none to 4—very severe). Skin hydration and structure, pH, transepidermal water loss, erythema, and melanin index were analyzed instrumentally. Creams 1, 2, and 3 significantly reduced the area of psoriatic plaques. Improvement in erythema and infiltration was observed for creams 1 and 3. Creams 1–3 reduced scaling. Our study confirmed a beneficial effect of creams containing *S. coronata* extract on psoriatic lesions.

## 1. Introduction

Psoriasis is a chronic, immune-mediated inflammatory skin disease, affecting approximately 2–3% of the global population, and remains one of the most commonly encountered conditions in dermatological practice [1]. The worldwide prevalence of psoriasis varies and is higher in wealthier countries and among older populations [2]. The pathogenesis of psoriasis is multifactorial, including genetic, environmental, and immunological factors [3]. One of the most common types, plaque psoriasis, presents with large, well-defined erythematous plaques, covered with silver-white scales, over the scalp, trunk, and extensor body surface, especially on the elbows, knees, and the lower back [4]. The Psoriasis Area and Severity Index [PASI] is used to monitor disease severity and effectiveness of the therapy [5,6]. Based on the PASI score, plaque psoriasis can be graded as mild, moderate, and severe [7]. Psoriasis is characterized by periods of remissions and exacerbations, the latter often requiring a prolonged therapy. Topical therapy and phototherapy are used in mild-to-moderate psoriasis, and systemic therapy is indicated for a more severe course of the disease. Understanding of the process of the disease is helpful in the development of novel and targeted therapies such as biologics [8]. Almost 80% of the affected patients present with mild to moderate psoriasis [4,9]. Depending on disease severity, topical treatment, phototherapy, photochemotherapy, and systemic treatment may be used [8,9,10]. Apart from topical and systemic pharmacotherapy, adequate skincare is necessary to maintain remission of the skin lesions [11]. Humectants, emollients, as well as keratolytic and anti-inflammatory agents are among the cosmetics whose beneficial effect on psoriatic skin has been confirmed [12]. The role of cosmetics for psoriatic skin is to maintain acidic pH [13]. Strengthening of the skin barrier aims to ensure adequate skin hydration and protection against water loss, which may lower the probability of developing the Koebner’s phenomenon [12,13,14]. The exfoliating action of the urea, salicylic as well as alpha- and poly-hydroxy acids is used for both therapy and skincare in patients with psoriasis [12]. A beneficial effect on psoriatic skin has been reported for numerous components which are found in various medicinal plants, e.g., aloe [15,16], hypericum [17], turmeric (*Curcuma longa* L.) [18], and indigo (*Baphicacanthus cusia* Brem.) [17,18,19]. Catechin, a green tea component, was found to reduce erythema, inflammatory infiltrate, and scaling on psoriasis-like skin lesions in mice [20]. Topical application of soybean protein on psoriatic model in mice inhibited epidermal weight, while genistein lowered the intensity of adverse effects associated with photochemotherapy (PUVA) [21,22]. Based on the available literature, it seems safe to assume that plants which exhibit anti-inflammatory properties and strengthen the skin barrier should have a beneficial effect on plaque-type psoriasis after topical use [12,16,18,21,22]. Due to their low toxicity and unique properties, phytoecdysones, which belong to the group of natural steroid compounds, seem to be promising agents. Phytoecdysteroids are found at various concentrations in the leaves and flowers of almost all plant species, including *Serratula coronata* L. (*S. coronata*). Apart from arbutin and flavonoids, phytoecdysteroids constitute the main group of chemical compounds found in that herb. Four main phytoecdysteroids have been identified: 20-hydroxyecdysone (20-HE), integristerone A, 22-deoxy-20-HE, and polypodine B [23]. Phytoecdysones exert a number of actions which improve the functioning of the skin. They are regulators of keratinocyte differentiation and restore skin hydration. These compounds strengthen the natural protective barrier of the skin, thus inhibiting the process of transepidermal water loss (TEWL). Phytoecdysones are especially recommended as ingredients of skincare products for dry and very dry skin, i.e., skin with disturbed keratinocyte differentiation. In addition, ecdysones affect the strength of the epidermis and improve epidermal activity leading to exfoliation, thus restoring the smoothness of the skin [24]. Particular attention is paid to the antioxidant and anti-inflammatory effects of these bioactive compounds, resulting in significant benefits in the treatment and prevention of human diseases [25].

The aim of the study was to determine the effect of cosmetic creams containing the extract of *Serratula coronata* on skin in patients with psoriatic lesions.

## 2. Results

### 2.1. Determination of Dominant Ecdysteroids in Formulations Using the ESI/HPLC-MS Method

The results of the qualitative analysis of creams 1 and 3 using the ESI/HPLC-MS were presented in the form of chromatograms (Figure 1) and mass spectra (Figure 2). Three phytoecdysteroids: 20-HE, ajugasterone C, and polypodine B were found.

### 2.2. Patch Tests

No allergic or irritant reactions were observed for creams 1 and 3, either in psoriatic patients or controls (Figure 3).

### 2.3. Results of the Clinical Assessment of Psoriatic Lesions

The results of the clinical assessment of the psoriatic lesions are presented in Table 1.

No statistically significant changes in lesion size were found for cream 0 (placebo), whereas creams 1–3 significantly reduced the surface of the psoriatic plaque (*p* = 0.0015, *p* = 0.0176, and *p* = 0.0179 respectively). The highest statistically significant improvement in erythema (*p* < 0.0001) was observed for creams 1 and 3. No difference in the reduction of erythema was found between these two creams. The most statistically significant improvement in plaque infiltration (*p* < 0.0001) was observed for creams 1 and 3. The most statistically significant improvement in scaling was observed for creams with 3% *S. coronata* extract and salicylic acid (*p* < 0.0001 creams 1–3). No difference in the reduction of scaling was observed between any of the investigated creams.

The psoriatic plaques, at baseline and at follow-up, from both investigated regions (knee, elbow) are presented in Figure 4.

### 2.4. Measurements of Skin Biophysical Parameters

No differences between the groups were found at baseline (*p* > 0.05). Data about skin biophysical parameters: melanin index (MI), erythema index (EI), pH, hydration, transepidermal water loss (TEWL) and their comparison at baseline and follow-up are presented in Table 2.

Skin structure improved after application of cream 1, from abnormal (13.17 at baseline) to normal (9.7 at follow-up). No changes in skin structure were observed for creams 0, 2, and 3. Skin structure images of psoriatic lesions after using cream 1 are presented in Figure 5.

## 3. Discussion

The available pharmacotherapies for psoriasis will help achieve remission but will not treat the disease. Therefore, patients with psoriasis often seek safe, natural skincare products and preparations to additionally support their treatment regimens [12,24,26], and medicinal plant-based cosmetics appear to offer such a chance to the affected individuals [27]. Randomized, controlled clinical trials have confirmed beneficial effects of certain medicinal plants—*Aloe vera, Centella asiatica, Panax ginseng, Rubia cordifolia, Saccharum officinarum*—on skin symptom alleviation in psoriasis [28,29,30,31,32,33,34,35,36]. In this study, we used the extract of *S. coronata*. The choice of the herb was based on the results of preliminary studies, which indicated biological activity of the phytoecdysteroids found in *S. coronata*. Their activity may improve the condition of the psoriatic lesions owing to their anti-inflammatory action [25] and normalization of keratinocyte differentiation [24,36].

Earlier studies confirmed 20-hydroxyecdysone, polypodine B, and ajugasterone C to be the dominant compounds of the *S. coronata* ecdysteroid fraction [37] and demonstrated the effectiveness of creams containing phytoecdysteroids aqueous extract of *S.coronata* in skincare for patients with seborrheic dermatitis [38]. Skin patch tests are one way of excluding the risk of allergic action of topical preparations and, consequently, confirming their safety. Natural products, including plant extracts, are often combinations of numerous chemical components, with varying allergenic potential, depending on their origin. That, in turn, makes all attempts at standardization challenging and may significantly affect the results of the patch tests [39]. For example, approximately 3000 compounds belonging to various classes of sesquiterpenoids have been identified in Asteraceae, with approximately half of them being potential contact allergens [40,41]. In this study, the patch tests revealed neither allergenic nor irritant reaction of the *S. coronata* extract creams in study group and controls.

It should be noted that the permeability of the active substances significantly affects the effectiveness of cosmetic formulations [42,43]. The effect of *S. coronata* creams on psoriatic lesions was assessed clinically and instrumentally. The measurements were used to determine the integrity of the epidermal barrier, whose functions are notably weakened in psoriatic lesions. The highest hydration activity was found in cream 0 (placebo) and creams 1 (*S. coronata* extract) and 3 (*S. coronata* extract with salicylic acid). Their hydration activity is most probably associated with the emollient activity of Lekobaza^®^, and the regenerative effect of the phytoecdysteroids on the skin barrier, which was confirmed by the TEWL results. Psoriatic lesions are associated with severe dryness so emollient therapy is necessary [44]. The choice of the base selected for our study has been validated by our results.

Skin pH significantly affects the skin barrier function and its overall condition. In this study, we investigated the effect of *S. coronata* creams on the pH of the psoriatic plaques and found that creams with salicylic acid statistically significantly lowered their pH. This is an important finding as physiological, slightly acidic, pH of the stratum corneum (ranging from 4.1 to 5.8) affects the barrier functions of the skin, synthesis, and aggregation of the lipids, as well as differentiation and exfoliation of the epidermal layer [13,45], not to mention composition of the microbiome and activation of the protective proteins. Microbiological colonization and interactions between the antigens and the enzymes of the micro-organisms and the skin immune system of the host may intensify and trigger psoriatic symptoms. Acidic pH inhibits colonization with pathogens such as *Staphylococcus aureus* or *Streptococcus pyogenes* and helps achieve normal microbial flora of the skin [46]. Inflammatory skin disorders are associated with higher pH, as compared to normal healthy skin. Higher pH and dry skin stimulate the proteinase-2-activated receptor, which leads to pruritus in patients with psoriasis as extra- and inter-cellular proton concentration modulates afferent (pruritus and pain) and efferent (growth, differentiation, and survival of cells) functions [45]. Lower skin pH results in higher antibacterial activity, change in protease activity, release of oxygen, decreased toxicity of the end products of bacteria, improved epithelialization, and angiogenesis [45]. Cosmetics which lower the skin pH, such as the creams used in our study, may exert a beneficial effect on psoriatic lesions, which was demonstrated by different authors [13,45,46]. A statistically significantly lower MI was confirmed in groups using creams with *S. coronata* and *S. coronata* with salicylic acid. Phytochemical analysis revealed the presence of flavonoids in *S. coronata* (quercetin, 3-methyloquercetin, apigenin, and isokaempferide), with antioxidative properties. Together with phytoecdysteroids, they alleviate the inflammatory response, thus lowering the risk for post-inflammatory hyperpigmentation. The brightening of the skin is also associated with the action of salicylic acid, beta hydroxy acid, which reduces keratinocyte proliferation, and of the stratum corneum [47]. No effect of the cream on EI was observed. That finding was not consistent with the clinical assessment of the erythema, which revealed lower intensity of erythema in all investigated lesions. It might have been the consequence of the exposure of the erythematous plaque after the scales had been removed from its surface. Reduced scaling was confirmed by tests as well as clinical assessment.

The clinical findings revealed significant reduction of the erythema, inflammatory infiltrate, and scaling of the psoriatic lesions after using the creams. The most significant reduction of the erythema was found for creams with *S. coronata*, which might be associated with the activity of phytoecdysteroids and flavonoids. Erythema reduction correlated with lower degree of scaling and plaque infiltration, which proves that the investigated creams help restore skin barrier. The creams with *S. coronata* and *S. coronata* with salicylic acid statistically significantly reduced the area of the psoriatic lesions. Spectral reflectance test, a non-invasive method to check the amount of light reflected on the skin surface, was used to assess the effect of the creams on skin texture [48]. Baseline measurements revealed abnormal skin structure in all analyzed psoriatic lesions, which is characteristic for psoriatic changes. After 6 weeks of using the creams, an improvement in skin structure was observed, which might be connected with their emollient activity. At the same time, that result emphasizes the importance of regular use of emollients in psoriatic skincare.

## 4. Materials and Methods

The study followed the Declaration of Helsinki and was approved by the Local Ethics Committee (Poznan University of Medical Sciences, No. 523/19, 11 April 2019, Poland). Written informed consent was obtained from all participants.

### 4.1. Material

The herbs of *S. coronata* were collected between June and July 2020 at the Botanical Garden of the Department and Division of Practical Cosmetology and Skin Diseases Prophylaxis, University of Medical Sciences in Poznan (Poland). The method of herb extraction and its phytochemical characterization was described in detail in our previous study [38]. The dried and ground sample in the amount of 2000 g was poured three times with 10, 250 mL of ethanol (96% CZDA; POCH, Gliwice, Poland). The resulting extract was filtered, and its solvent was evaporated using a rotary evaporator (Rotavapor^®^ R-100, BUCHI, Flawil, Switzerland) at 40 °C. The extract was concentrated to dryness at room temperature. A total of 136.42 g of the extract (*extractum siccum*) in dark green color was obtained. For the purpose of the study, a total of 80 creams, with 4 different formulas (0–3) and a capacity of 30 g, were prepared. The composition of the creams is presented in Table 3.

Lekobaza^®^ Pharma Cosmetic is a multi-component medium with a pH of 5.5 (Fagron, Kraków, Poland), consisting of white Vaseline, glycerol monostearate, and cetyl alcohol, Miglyol^®^ 812, macrogol-20-glycerol-monostearate, propylene glycol, and purified water. It was used as the cream base and as placebo. It is a universal base which, like other o/w emulsions, easily penetrates the skin, is less greasy, and creates a delicate protective film on the skin surface.

The ingredients of creams 1–3 were combined by grinding in a mortar. Then, they were transferred to plastic pharmacy boxes (50 mL) and marked with consecutive numbers (0–3).

### 4.2. Study Group

A randomized, double-blind, placebo-control prospective study was conducted in a group of 70 patients with plaque-type psoriasis (34 men and 36 women, aged ≥18 and ≤70 years). The participants were recruited among patients of dermatological out-patient clinics. The inclusion criteria were as follows: no history of systemic therapy for psoriasis, plaque psoriasis with PASI < 7, Body Surface Area (BSA) ≤ 3, Dermatology life Quality Index (DLQI) < 10, lesions localized on elbows or knees, consent to topical treatment, photo- and heliotherapy not being used over the course of the study. The patients consented that topical treatment would be used only for the investigated creams on selected elbow- or knee-located psoriatic lesions. The exclusion criteria included: allergy, neoplastic diseases, autoimmune diseases, pregnancy and breastfeeding, use of aggressive general medications, and positive patch test results. Ten additional healthy volunteers were subjected to the patch tests with the investigated creams and constituted the control group. Detailed characteristics of the study group are presented in Table 4.

### 4.3. Methods

#### 4.3.1. Determination of Dominant Ecdysteroids in Creams Using the ESI/HPLC-MS Method

Qualitative analysis using ESI/HPLC-MS was applied to confirm the presence of the dominant phytoecdysteroids (20-HE, ajugasterone C, and polypodine B) in the creams. In order to determine the influence of the atmospheric conditions (temperature, humidity, and UV radiation) on the stability of these compounds, the creams which had been tested in a climatic chamber were also submitted for analysis. The obtained samples were diluted 1000-fold before the analysis. The measurements were performed using a QTOF mass spectrometer (Impact HD, Bruker Daltonics, Billerica, MA, USA) in positive ion mode and an Ultimate 3000 liquid chromatograph (Thermo Scientific/Dionex, Waltham, MA, USA). A Kinetex column (2.6u C18 (100 × 210 mm) was eluted with water with 0.1% formic acid (A) (Sigma Aldrich, Saint Louis, MO, USA) and acetonitrile (Sigma Aldrich, Saint Louis, MO, USA) with 0.1% acetic acid (B) at a flow rate of 0.3 mL/min. Initially, B was held at 10% for 1 min, and then was increased linearly to 90% B at 24 min and held for 2 min. The mobile phase was then returned to the initial condition.

#### 4.3.2. Clinical Assessment

All participants were examined by a dermatologist to assess disease severity with the use of PASI [5,6,7], BSA, and DLQI. BSA measures the percentage of the body covered with psoriatic lesions. The hand rule, i.e., that the area equivalent to one hand of the patient corresponds to 1% of their body surface, was used to determine the extent of psoriasis. The result was obtained by adding up the percentage of skin lesions, and classified as mild (BSA < 3%), moderate (BSA 3–10%), or severe (BSA > 10%) [49]. The Polish version of DLQI was used at baseline to assess the impact of psoriasis on patient quality of life. A higher score is associated with a more negative impact of psoriasis on quality of life [50,51]. The combination of PASI, BSA, and DLQI scores allows to determine the level of disease severity. Only individuals with mild psoriasis were included in the study.

Assessment of the 72 psoriatic plaques located on the elbows (20) and the knees (52) was based on the PASI score [5,6]. The designated cream was applied twice daily (the amount depended on the size of the lesions with the use of a finger-tip unit—FTU) [52] for 6 weeks. The investigated area was measured, and severity of scaling, erythema, and infiltration of the psoriatic plaque was assessed using a 5-point scale (0—none, 1—mild, 2—moderate, 3—severe, 4—very severe). Lesion assessment was performed at baseline and at follow-up, 6 weeks later.

#### 4.3.3. Patch Tests

Patch tests (PTs) were performed to determine biosafety of creams 1 and 3. Both in the study group (10 patients) and the control group (10 healthy volunteers), PTs were conducted at the Department of Dermatology, Poznan University of Medical Sciences. PTs were applied using Finn Chambers mounted on intact skin of the interscapular area for 48 h. PT readings were performed 48 and 72 h after application. The results were interpreted in accordance with the guidelines of the International Contact Dermatitis Research Group (ICDRG) [53].

#### 4.3.4. Measurement of Skin Biophysical Parameters

The effect of the creams on psoriatic skin was determined by measuring the biophysical parameters. Skin hydration measured with corneometry, pH, TEWL, intensity of erythema, melanin and skin structure were analyzed. A series of 10 measurements was performed for each parameter, and the final result was presented as mean value. The measurements were performed twice: at baseline and after study completion. On testing day, the creams were not used at least 1 h before the measurements. To avoid bias, all tests were performed in the same room, at a temperature of 22 ± 3 °C and air humidity of 45 ± 2%. Two skin analyzers were used—Courage & Khazaka electronic (GmbH, Cologne, Germany) and the Nati Skin Analyzer device (Beauty of Science, Wroclaw, Poland). The following probes were used in the study: Corneometer^®^ CM 825 (Courage & Khazaka, Koln, Germany)—to analyze skin hydration; Tewameter^®^ TM 300 (Courage & Khazaka, Koln, Germany)—to assess skin barrier function by measuring transepidermal water loss; Mexameter^®^ MX 18 (Courage & Khazaka, Koln, Germany) to assess melanin index (MI) and erythema index (EI); and Skin-pH-Meter PH 905 (Courage & Khazaka, Koln, Germany). The NATI Skin Analyzer is a device designed for cosmetological diagnostics of the skin. Its measurement system is based on physical and optical analysis. The use of two different light sources (UV and white light) allows for proper classification of a given skin parameter. NATI is equipped with a modern digital camera using HD Ready technology. High-quality photographic documentation may be obtained due to the possibility of 40x enlargement of the image. The following parameters are analyzed during the examination: skin structure, lubrication, exfoliation, skin pore diameter, wrinkle width, vascular changes, discoloration, and the degree of hydration. The values of the measured parameters are presented using a color scale. Color saturation (red, yellow, green) defines the quality of the obtained result: green—very good result, yellow—intervention is required, red—alarming. The results are interpreted based on the color range where the skin structure is located: 0–8 excellent (green area); 8–10 good (yellow area); >10 incorrect (red area).

#### 4.3.5. Statistical Analysis

Statistical analysis was based on TIBCO Software Inc. (2017). Statistica (data analysis software system), version 13 (Tulsa, OK, USA). All results were first verified by a normality test (Shapiro–Wilk test). Since the test confirmed a lack of normality, non-parametric methods were used for statistical analysis. Differences between before and after creams application were tested by the Wilcoxon test. Differences between groups results were verified by the Kruskal–Wallis test. The *p*-value of <0.05 was considered as statistically significant.

## 5. Conclusions

Our findings confirmed a beneficial effect of creams containing extract of *S. coronata* on psoriatic lesions. Therefore, it seems safe to conclude that it is a promising component of skincare products for psoriatic patients.

## Figures and Tables

**Figure 1 molecules-27-03471-f001:**
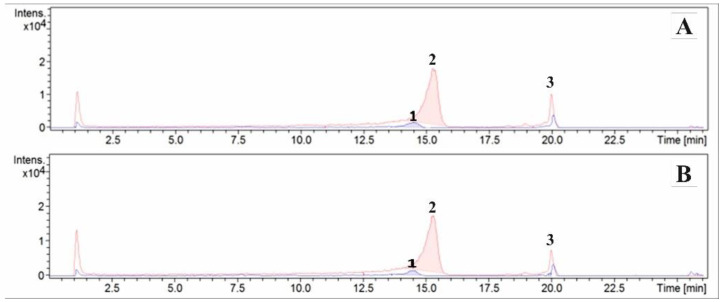
Chromatogram of cream 1 (**A**) and cream 3 (**B**) containing 3 wt.% *S. coronata*: (1) polypodine B; (2) 20-hydroxyecdysone; (3) ajugasterone C.

**Figure 2 molecules-27-03471-f002:**
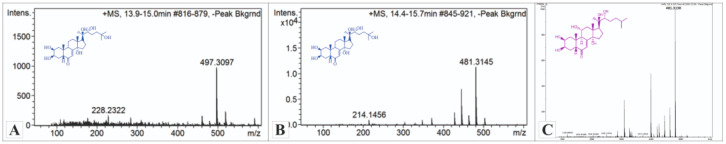
Polypodine B (**A**), 20-hydroxyecdysone (**B**), ajugasterone C (**C**) in 3 wt.% *S. coronata* creams identified with ESI-MS.

**Figure 3 molecules-27-03471-f003:**
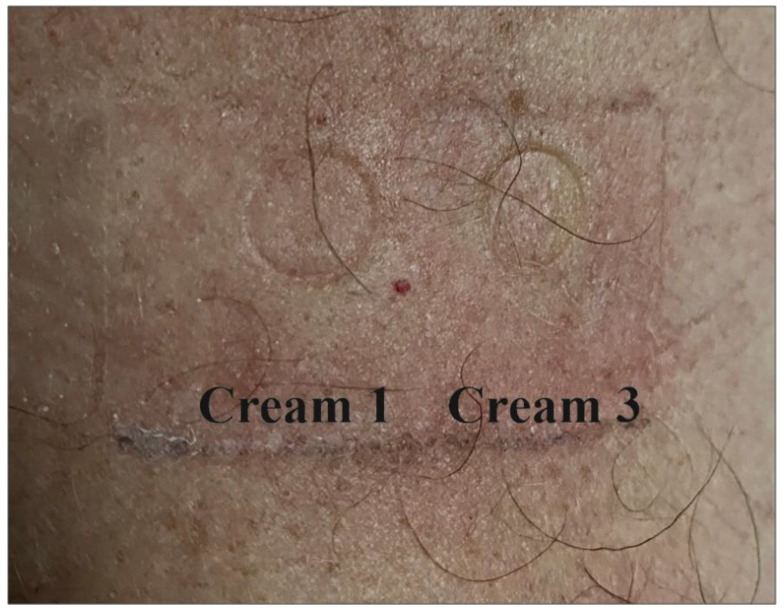
Negative patch testing results for the cream 1 and cream 3 in a psoriatic patients.

**Figure 4 molecules-27-03471-f004:**
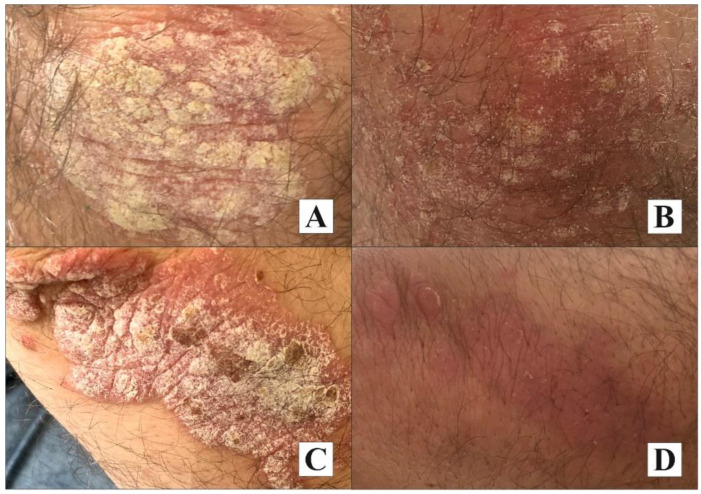
The psoriatic plaques at baseline (**A**,**C**) and at follow-up (**B**,**D**) (cream 1).

**Figure 5 molecules-27-03471-f005:**
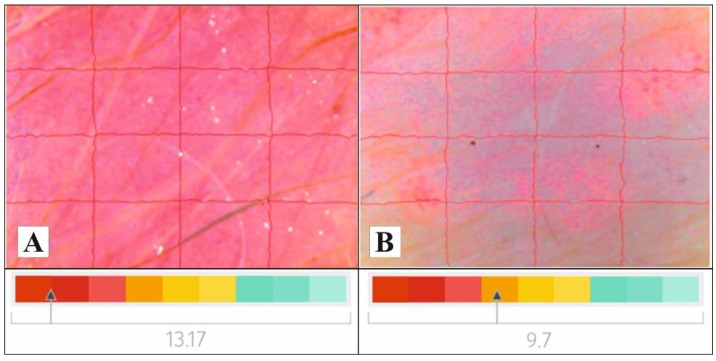
Images of psoriatic lesion skin structure (**A**) before and (**B**) after application of cream with *S. coronata*). Results were expressed as a reading scale of colors: green—really good structure; yellow—normal skin structure; red—disordered skin structure.

**Table 1 molecules-27-03471-t001:** Results of the clinical assessment of psoriatic lesions at baseline and at follow-up.

Characteristics of Psoriatic Lesions		Cream 0 (n = 17)	Cream 1 (n = 20)	Cream 2 (n = 18)	Cream 3 (n = 15)
**Area (cm^2^)**	Baseline mean ± SD	14.8 ± 21.2	17.4 ± 22.7	20.1 ± 33.1	14.1 ± 14.2
	follow-up mean ± SD	14.6 ± 20.8	14.6 ± 20.1	19.0 ± 32.8	12.4 ± 12.8
	*p* value	0.0679	0.0015 *	0.0176 *	0.0179 *
**Erythema (points)**	Baseline M(range)	2 (1–4)	2 (1–3)	2 (1–4)	2 (1–4)
	follow-up M (range)	1 (1–4)	1 (1–3)	1 (0–4)	1 (0–3)
	*p* value	0.0175 *	<0.0001 *	0.0032 *	<0.0001 *
**Infiltration (points)**	Baseline M (range)	1 (0–3)	2 (0–4)	1 (0–3)	2 (1–4)
	follow-up M (range)	0 (0–3)	0.5 (0–2)	0 (0–1)	0 (0–2)
	*p* value	0.0336	<0.0001 *	0.0022	<0.0001 *
**Scaling (points)**	Baseline M (range)	1 (0–2)	2 (0–3)	1 (0–3)	2 (0–3)
	follow-up M (range)	1 (0–2)	1 (0–2)	0.5 (0–2)	1 (0–3)
	*p* value	>0.05	<0.0001 *	<0.0001 *	<0.0001 *

Descriptions: n—number of psoriatic lesions; SD—standard deviation; M—median; * statistically significant result.

**Table 2 molecules-27-03471-t002:** Detailed results and comparison of skin biophysical parameters at baseline and follow-up.

Parameter		Cream 0 (n = 17)	Cream 1 (n = 20)	Cream 2 (n = 18)	Cream 3 (n = 15)	Comparison between Creams (*p*)
**Melanin** **Index (AU)**	Baseline Mean ± SD	148.5 ± 79.9	164.5 ± 79.8	150.0 ± 37.8	199.8 ± 88.8	0.1576
	follow-up Mean ± SD	141.1 ± 57.8	150.9 ± 65.9	127.2 ± 28.0	150.8 ± 67.9	0.6286
	*p* value	0.9811	0.1259	0.0057 *	0.0007 *	
**Erythema Index (AU)**	Baseline Mean ± SD	409.9 ±113.2	407.8 ± 108.6	402.4 ± 89.1	409.4 ± 105.3	0.9870
	follow-up Mean ± SD	435.2 ± 74.7	437.3 ± 89.0	433.2 ± 76.3	456.4 ± 106.2	0.6611
	*p* value	0.0684	0.4115	0.0854	0.1118	
**pH**	Baseline Mean ± SD	5.54 ± 0.52	5.79 ± 0.81	5.77 ± 0.49	5.9 ± 0.9	0.6953
	follow-up Mean ± SD	5.56 ± 0.51	5.72 ± 0.78	5.54 ± 0.81	5.25 ± 0.47	0.1184
	*p* value	0.5862	0.5862	0.0166 *	0.0066 *	
**Hydration** **(AU)**	Baseline Mean ± SD	12.5 ± 7.8	10.2 ± 7.6	13.1 ± 11.6	12.1 ± 6.9	0.7626
	follow-up Mean ± SD	17.6 ± 8.5	19.6 ± 8.6	15.9 ± 8.5	18.5 ± 8.2	0.2016
	*p* value	0.0003 *	0.0001 *	0.0936	0.0076 *	
**TEWL (g/h/m^2^)**	Baseline Mean ± SD	23.1 ± 9.8	21.4 ± 12.1	21.4 ± 8.8	21.7 ± 9.6	0.8672
	follow-up Mean ± SD	16.9 ± 8.8	14.4 ± 6.6	17.8 ± 6.2	16.4 ± 9.3	0.3453
	*p* value	0.0003 *	0.0001 *	0.0582	0.0108 *	

Descriptions: n—number of psoriatic lesions; SD—standard deviation; AU—arbitrary unit; *—statistically significant result.

**Table 3 molecules-27-03471-t003:** Cream composition.

Creams	Composition	Ingredient Content (g)
**0**	Lekobaza^®^ (placebo)	30
**1**	Lekobaza^®^/*S. coronata* extract	27/3
**2**	Lekobaza^®^/salicylic acid	25/5
**3**	Lekobaza^®^/*S. coronata* extract/salicylic acid	22/3/5

**Table 4 molecules-27-03471-t004:** Patient characteristics.

Parameter (Mean ± SD)	All (n = 70)	Female (n = 36)	Male (n = 34)
**Age [years]**	41.1 ± 13.1	41.1 ± 13.2	41.1 ± 13.2
**PASI [points]**	4.2 ± 2.9	3.0 ± 1.8	5.4 ± 3.4
**BSA [%]**	2.7 ± 0.5	2.8 ± 0.4	2.7 ± 0.4
**DLQI [points]**	5.2 ± 3.2	5.2 ± 3.3	5.2 ± 3.0

Descriptions: n—number of patients; SD—standard deviation PASI—psoriasis area and severity index; BSA—body surface area; DLQI—dermatology life quality index.

## Data Availability

Not applicable.

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
