# Peer review of "Phytoecdysteroids from Serratula coronata L. for Psoriatic Skincare"

_molecules, 2022, doi:10.3390/molecules27113471_

Round 1
Reviewer 1 Report
The manuscript molecules-1744785 presents a meaningful research work with a clear process and results with potential clinical application. Overall there are no major problems with the manuscript, some details such as the formatting of references (pagination of references 4, 10, 52 etc.) need to be corrected.
No problems have been identified in the detail of the manuscript, except for a few flaws in the references. In addition, it is clear that Figure 5 is too crude for a scientific paper that seeks aesthetics based on objective and accurate illustrations. Figure 5 creates a strong visual impact and may cause discomfort to some readers. It is recommended that the redundant parts be cropped out and only specific parts be featured, with the missing information being described in text. As I have cropped the image in the attachment, it is for reference only.

Reviewer 2 Report
The manuscript is generally well written and clearly presented. It studied the effects of phytoecdysteroids on psoriatic lesions.
Comments:
1. Line 327: The method for permeability should be optimized. For example, the authors incubated skin at 37 °C. However, 37 °C is normally the temperature for the receptor medium. The skin surface temperature should maintain at 32 °C. In addition, high humidity in the incubator can also increase skin hydration. The conditions used by the authors can overestimate skin permeation.
2. The authors may need to consider removing section 2.2 and the discussion of permeability because the experimental conditions are inappropriate. Details of data processing for Raman spectroscopy should also be given if the authors want to keep this section.
3. Section 2.3 The authors could provide some photos to show the results of patch tests.
4. Line 138: please give full terms of MI, EI, TEWL
Round 2
Reviewer 1 Report
The author has made careful revisions to the manuscript to improve the quality of the manuscript, and it is recommended to accept this manuscript.